# Flexible Microstructured Capacitive Pressure Sensors Using Laser Engraving and Graphitization from Natural Wood

**DOI:** 10.3390/molecules28145339

**Published:** 2023-07-11

**Authors:** Chenkai Qu, Meilan Lu, Ziyan Zhang, Shangbi Chen, Dewen Liu, Dawei Zhang, Jing Wang, Bin Sheng

**Affiliations:** 1School of Optical Electrical and Computer Engineering, University of Shanghai for Science and Technology, Shanghai 200093, China; 1935031916@st.usst.edu.cn (C.Q.); 1935030804@st.usst.edu.cn (M.L.); 1935030808@st.usst.edu.cn (Z.Z.); dwzhang@usst.edu.cn (D.Z.); 2Shanghai Key Laboratory of Modern Optical Systems, Engineering Research Center of Optical Instruments and Systems, Shanghai 200093, China; 3Inertial Technology Division, Shanghai Aerospace Control Technology Institute, Shanghai 201109, China; chensb@mail.ustc.edu.cn (S.C.); dewerl@126.com (D.L.); 4Department of Ultrasound Medicine, The First Affiliated Hospital, Zhejiang University School of Medicine, Hangzhou 310006, China; yanshu006@zju.edu.cn

**Keywords:** laser-induced graphene, flexible pressure sensor, microstructures, capacitance sensors, laser engraving

## Abstract

In recent years, laser engraving has received widespread attention as a convenient, efficient, and programmable method which has enabled high-quality porous graphene to be obtained from various precursors. Laser engraving is often used to fabricate the dielectric layer with a microstructure for capacitive pressure sensors; however, the usual choice of electrodes remains poorly flexible metal electrodes, which greatly limit the overall flexibility of the sensors. In this work, we propose a flexible capacitive pressure sensor made entirely of thermoplastic polyurethane (TPU) and laser-induced graphene (LIG) derived from wood. The capacitive pressure sensor consisted of a flexible LIG/TPU electrode (LTE), an LIG/TPU electrode with a microhole array, and a dielectric layer of TPU with microcone array molded from a laser-engraved hole array on wood, which provided high sensitivity (0.11 kPa^−1^), an ultrawide pressure detection range (20 Pa to 1.4 MPa), a fast response (~300 ms), and good stability (>4000 cycles, at 0–35 kPa). We believe that our research makes a significant contribution to the literature, because the easy availability of the materials derived from wood and the overall consistent flexibility meet the requirements of flexible electronic devices.

## 1. Introduction

Flexible pressure sensors are increasingly being paid more attention, because their flexibility, light weight, and stretching are crucial to a wide range of applications in surgical operation [1,2,3,4], moisture detection [5,6,7,8,9,10], human motion [11,12,13,14], and wearable devices [15,16,17], etc. According to various working mechanisms, flexible pressure sensors can be categorized by resistivity [10,18,19], capacitance [20,21,22], triboelectricity [23], and piezoelectricity [24,25]. Among them, capacitive flexible sensors have been studied extensively by researchers because of their simple structure, easy assembly, and wide range of applications [20,21,22,26]. A large number of microstructures comprising micropyramids, microdomes, semicylinders, porous structures, etc., have been applied for improving the sensitivity of capacitive pressure sensors [20,22,26,27,28,29,30,31,32]. Since most microstructures are manufactured using processes such as photolithography, PVC, etc., their processes are complicated, expensive, and time-consuming [28,30]. As an emerging method, laser engraving has received a great deal of attention for fabricating microstructures because of its higher accuracy, simpler process, and greater convenience compared to traditional methods [27,33].

In addition to processing methods such as fabricating microstructure, laser engraving is also able to fabricate laser-induced graphene (LIG) on diverse precursors ranging from polymers to renewable precursors such as cloth, paper, and woods, etc. [18,34,35,36,37,38,39]. LIG, as an excellent porous 3D graphene-based nanomaterial, is easily obtained from materials such as polyimide (PI), polydimethylsiloxane (PDMS), or polyurethane for strain, temperature, and humidity detection [36,40,41,42,43,44]. LIG transferring with elastic materials has also shown a high performance in supercapacitors, physical/chemical sensors, etc. For instance, Hong et al. prepared durable and permeable strain sensors by transferring LIG to nose film colloid [45]. Mahmood et al. transferred LIG from the kraft lignin film onto polydimethylsiloxane (PDMS) for supercapacitors [46]. Additionally, as a subtractive manufacturing process that enables the rapid prototyping of flexible sensors, a laser was used to engrave acrylic molds to obtain a very sharp microcone.

Although previous studies have prepared graphene via laser and then used it as a conductive substance for resistive sensors, or have used the laser as a preparation method for microstructured conductive layers, the use of a laser to form a conductive substance while performing microstructure processing has not been studied, which smartly takes advantage of the fact that wood can be used as both a microstructured template and a source for conductive substance, as shown in our manuscript. The characteristics of this process allow the dielectric layer to be integrated with the electrode, which effectively unifies the flexibility of the electrode and the dielectric layer.

In this work, we successfully realized the integration of the LIG electrode and LIG electrode with a laser-engraved microstructure via secondary laser engraving on wood and peeling it off using thermoplastic polyurethane (TPU). We demonstrated that minimizing the ablative effect can improve the amount of LIG generated by pretreating the wood with a flame-retardant treatment in order to make the resistance of LIG generated by laser scribe modest enough. Using laser engraving, we also produced a microcone structure on an LIG electrode rapidly and simply, which significantly increased the sensor’s sensitivity. The capacitive pressure sensor consisted of a flexible LIG/TPU electrode (LTE), an LIG/TPU electrode with a microhole array, and a dielectric layer of TPU with a microcone array molded from a laser-engraved hole array on wood, which provided high sensitivity (0.11 kPa^−1^), an ultrawide pressure detection range (20 Pa to 1.4 MPa), a fast response (~300 ms), and good stability (>4000 cycles, at 0–35 kPa). This method solves the flexibility problem caused by the electrode, and it is more economical, durable, and less time-consuming for application.

## 2. Results and Discussion

### 2.1. Laser Operation Parameter Optimization of the LTE

The laser scribing of natural materials such as wood and leaves can produce LIG with a good performance, as is known from previous research [37,38,39], and the LIG produced by planks treated with flame retardant has significantly better resistance than that of untreated ones, as will be demonstrated later. In order to find the most suitable parameters for LTE, we first conducted a comprehensive study of LTE in terms of laser power, scan speed, and defocus distance. From the tests of laser power and scan speed (Figure 1a), we obtained a minimum sheet resistance (14.7 Ω/sq) of LIG with a 48 mm s^−1^ scan speed and 50% laser power without considering defocus. The defocus distance based on the above data was adjusted to obtain the smallest sheet resistance LIG at ~0.5 mm (Figure 1b). The error bars in Figure 1b represent the fluctuation of the eleven sets of experimental data, and the confidence interval was calculated by Origin automatically based on the standard deviation in the case of the certain defocus distance, 48 mm s^−1^ scan speed, and 50% laser power. It can be seen from the Figure that the difference in LIG with the same parameters obtained at a defocus distance less than 0.2 mm was larger. In order to further study the optimal sheet resistance for the parameters of scan speed, laser power, and defocus distance without considering the distance from the focus, we also studied the sheet resistance for different laser powers at a fixed scan speed (48 mm s^−1^) and defocus distance (~0.5 mm), and it can be seen that the sheet resistance was also minimum when the parameters of the lowest sheet resistance were taken for all three parameters (Figure 1a). The lowest sheet resistance of LIG was 14.7 Ω/sq. Therefore, we chose 50% laser power, 48 mm s^−1^ scan speed, and a 0.5 mm defocus distance to fabricate the LTE (the LTE and the LIG/TPU electrode with microhole array mentioned hereafter in this paper are prepared using this parameter). In addition, as we can see from Figure 1c, the sheet resistance increased significantly after incorporating TPU. This is because TPU is an insulator, and the infiltration of TPU leads to the destruction of some of the original conductive paths of LIG, so the sheet resistance increases. The error bars in Figure 1c represent the fluctuation degree of sheet resistance with different laser powers when the scan speed is 48 mm s^−1^ and the defocus distance is 0.5 mm.

Compared to the uniform LTE, the LIG/TPU electrode with microhole array was further considered. We measured the samples’ sheet resistance by a four-point prober with a 1 mm interval, so when the flexible electrode with microstructure was measured, the holes in the electrode were avoided as much as possible to obtain the minimum sheet resistance. The LIG/TPU electrode with microhole array could be calculated by the area ratio between the uniform LIG/TPU electrode without holes. Because the microhole array for the flexible electrode with microstructure had a 1 mm pitch with 0.5 mm diameter, we could find the ratio of the area of the holes on the electrode at about pi/16, i.e., 19.6%. The effective area of the electrode is about 80.4%, so the surface resistance of the LIG/TPU electrode with the microhole array should be the surface resistance of uniform LIG/TPU electrode/80.4%. During the four-probe test, the tip of the needle avoided the hole as much as possible, and it was found that the actual value of ~80% was close to the above theoretical value.

### 2.2. Characterization of LTE from Wood and the Effect of Flame Retardant

To further investigate the composition of LTE and the effect of flame retardant on the wood for LIG, chemical analyses of the surfaces of LTE and TPU were performed. Raman spectra were measured with Raman spectroscopy equipped with a 532 nm laser and a 785 nm laser. LIG could be measured clearly at 532 nm, but the characteristic peak of wood at 532 nm was extremely insignificant, which was due to the strong fluorescence interference of wood chips, so we changed the wavelength to 785 nm to reduce the fluorescence interference and enhance the signal-to-noise ratio of the Raman signal [47]. Three distinct peaks in the Raman spectrum were visible at 1326 cm^−1^, 1577 cm^−1^, and 2650 cm^−1^, corresponding to the D, G, and 2D bands, respectively, which could show the possible graphite structure [38]. The 2D peak was not obvious, which indicated that LIG was multilayer graphene. Last but not least, the momentum conservation achieved by two phonons with opposite wavevectors was what caused the 2D peak at about 2700 cm^−1^ to be activated without the need for any defects. As shown in Figure 2b–e, X-ray photoelectron spectroscopy (XPS) was performed to evaluate the detailed element information (Figure 2b). From Figure 2b, it can be seen that LIG (flame retardant) and LIG (not flame retardant) showed two peaks, in which the carbon 1s peak occurred at 285 eV, while the oxygen 1s peak occurred at 532 eV. For the LTE mixed with flame-retardant-treated LIG and non-flame-retardant-treated LIG and TPU, the atomic ratios of carbon 1s to oxygen 1s were 1.29 and 2.08, respectively. This result indicates that the ratio of carbon atoms to oxygen atoms decreased after flame-retardant treatment, which was contrary to the expected effect of flame-retardant treatment. Therefore, the C1s spectra of the two LIG samples with and without flame-retardant treatment were further investigated. In addition, no nitrogen or phosphorus was detected in the LIG (not flame retardant), while the LIG (flame retardant) contained 2.55% nitrogen and 9.93% phosphorus. This is because flame retardants contain a considerable amount of nitrogen and phosphorus elements. Figure 2c,d demonstrate that the C1s spectra of two samples of LIG, with flame retardant and without flame retardant, contained four peaks, located at 284.7 eV (C–C/C=C), 285.5 eV (C–N), 286.0 eV (C–O), and 289.0 eV (C=O), respectively [18]. It can be clearly seen that the content of the C–C/C=C bond increased from 40.9% to 51.7% with flame-retardant treatment, while the intensities of those from oxygen-containing groups (C–O, C=O) plummeted from 53.0% to 27.1%, which indicates that the flame-retardant treatment was effective. In addition, the content of the C–N bond increased from 6.0% to 21.0%. This was mainly because the flame retardant contained N and P elements, and the amount of N and P elements in the generated LIG was increased.

The thermogravimetric curves of wood and the wood with flame retardant are shown in Figure 2e, indicating that the decomposition temperature of the two samples was approximately 80 °C. With the heating process beyond 350 °C, the wood without flame retardant displayed the faster weight loss. This indicated that ablation produces more gas, leading to a reduction in char. The remaining weight of the wood with flame retardant declared that less ablation leads to more formation of char, which was consistent with research into flame-retardant nitrogen and phosphorus that has shown that the flame retardant properties of nitrogen and phosphorus decompose with heat to release nitrogen gas to isolate oxygen for a flame-retardant effect, producing more char in the process [48]. The above results all prove that flame retardants are effective in avoiding ablation and are favorable for the greater formation of LIG to reduce the LTE’s sheet resistance (Figure 2f). The error bars in Figure 2f represent the fluctuation degree of the sheet resistance of LIG between flame retardant and no flame retardant using different laser power.

### 2.3. Morphology and Sensing Mechanism of the LIG/TPU Electrode with Microhole Array, and Dielectric Layer of TPU with Microcone Array

The detailed morphology and microstructure of the LTE and the microcone of TPU are shown in Figure 3a–d. The microcone of TPU was formed by secondary laser engraving on the basis of the LTE. We applied a scanning electron microscope to investigate the surface morphology of the LTE and the microcone of the TPU. The surface of the LTE (Figure 3a) had a large linear structure, which can be seen in the magnified LTE surface (Figure 3b). This was mainly due to the directional nature of the fiber structure of the wood itself and the path that the laser continuously engraved through, while the few fine hairy structures on the LTE surface may have been the result of weak ablation or takedown damage. Figure 3c,d show the conical multilevel microstructure formed on the microcone of the TPU by a single laser pulse. On the basis of the cone, there were multilevel burr-like structures at different heights, and their direction was the same as the growth direction of the wood fibers, which was caused by the ablation that occurred inside the wood without flame-retardant treatment [49].

The formation of this microstructure was closely related to the laser power. The laser’s power distribution is shown in Figure 3e, and it can be observed that it had a Gaussian distribution [50]. I_ω_ was the minimum power at which the laser could transform the wood, and it can be seen that the impact area was the power range greater than I_ω_, and the transformation depth was proportional to the power of the laser; the higher the power, the deeper the transformation depth. The intensity cross section of the laser obeyed a Gaussian distribution, which was an important reason for the formation of microcones. As the laser ablation of the board proceeded, the gradual decoking led to a gradual reduction in the radius of the critical point, where the laser caused ablation and eventually the formation of the microcone.

After studying the formation principle of laser engraving microstructures, we further investigated the effect of laser engraving microstructures on pressure sensing performance. The correlation between laser engraving microstructures and pressure sensing performance was the height and radius of the microcone and the duty cycle of the microcone arrangement. The higher the height and smaller the radius of the microcone, the easier the microcone could compress and the higher the sensitivity of the sensor. We could adjust the height of the microcone by controlling the laser power and engraving time, i.e., scan speed. It can be seen from Figure 3f that the height of the microstructure formed from 200 μm to 1 mm. The radius of the microcone was determined by the effective ablation radius of the laser spot, which could be adjusted by controlling the defocusing distance, so we prepared the microcone array so that the defocusing distance was 0 and the laser spot was minimized. The duty cycle influenced the compressibility and capacitance of the sensors. The smaller the duty cycle of the microcone array, the higher the sensitivity of the sensor, but the initial capacitance of the sensor will slightly decrease [32,51]. The duty cycle of the microcone array was adjustable by computer. Therefore, after determining the appropriate microcone height and radius (at 55% laser power, 48 mm s^−1^, and without defocus), we chose the duty cycle of 1:2, and the final maximum sensitivity of the sensor was 0.11 kPa^−1^, with an initial capacitance of 4–6 pF.

For capacitive pressure-sensing applications, the unexpected stray capacitance caused by a capacitive effect between wires and components would be an important issue, so we measured the capacitive signals using a standard testing method, including shielding the RF signal from the testing environment with a relatively constant temperature, to keep the acceptable signal-to-interference-plus-noise ratio (>20). In order to research the stability and reproducibility of the laser engraving microstructure, we chose 55% laser power and 0.5 s engraving time to study the relative error of the microcone and prepared five sets of microcones under this condition parameter. Experiments showed that the bottom diameter of the microcone obtained with this parameter was almost 500 μm ± 20 μm, but the average heights of the microcones were 402 μm, 385 μm, 379 μm, 436 μm, and 453 μm, respectively. The height of the microcone at this parameter could be calculated from the experimental data to be ~400 μm, with a relative error of about 10%. The reason for this error may be the fluctuation in the laser power and error in focusing. Firstly, the maximum power of the 450 nm laser we used was 5 W, with fluctuations of 0.2 W, and the corresponding error was about 2%. Secondly, the error of focusing was attributed to the adjusting accuracy of the laser instrument and the nonparallel focal plane and wood surface, and this error of focusing was about 10%.

Figure 3g illustrates the sensing mechanism of the LMPS, which is a combination of the TPU dielectric layer and the LIG/TPU electrode, to form a complete capacitive pressure sensor. When the electrode surface is subjected to pressure, the microcone will be deformed, and the distance between the electrode and the electrode will be reduced to produce the change in capacitance. The burr-like structure on the cone had a small effect on the sensitivity, but it increased the contact area between the LTE and the microcone to a certain extent, which improved the stability between the two electrodes. The upper tip of the microcone is characteristically easily deformed, so while the lower bottom was wide with less deformation under the same pressure, the sensitivity of the sensor was high at low pressure, and when the distance between electrodes decreased and the ratio of air between the electrodes decreased, the dielectric constant between the electrodes increased and the TPU ratio increased two factors which affect the capacitance increases. The ratio of air between electrodes still decreased as the pressure kept increasing. However, the distance between the electrodes’ decreases became challenging, so the sensitivity decreased. When the pressure increased again, the air gap between the electrodes completely disappeared and the dielectric constant between the electrodes no longer increased, making the distance between the electrodes more and more difficult to reduce, so the sensitivity was even lower, which was consistent with the load curve shown in Figure 4a. In order to verify the above mechanism, we prepared the same amount of TPU into a flat dielectric layer and a dielectric layer with a microcone to measure its capacitance, and it can be seen from Figure 3h that the sensitivity was higher when the pressure was lower, because the microcone was more likely to deform, and when the air gap between the electrodes was gradually filled, the capacitance change was exactly the same as that of the flat TPU layer.

### 2.4. Electrical Properties of the LMPS Composed of LTE, LIG/TPU Electrode with Microhole Array and Dielectric Layer of TPU with Microcone Array

The LMPS consisted of an LTE, an LIG/TPU electrode with microhole array, and a dielectric layer of TPU with a microcone array. Due to the outstanding electrical conductivity of LTE, it can be used directly as an electrode and has good reliability. The electrical signal of the device changes under pressure are shown in Figure 4a. The LMPS had a wide detection window of up to 1393 kPa and the minimum detection limit of ~20 Pa. The electrical response of the device (Figure 4a) can be divided into three parts with different sensitivities at different load ranges. In the loading range of 0.01~20 kPa, the device had a sensitivity of ~0.11 kPa^−1^, and the sensitivity decreased to ~0.014 kPa^−1^ from 20 to 155 kPa. While the loading range was 155~1393 kPa, the sensitivity was ~0.003 kPa^−1^. This result shows that the LMPS had a high sensitivity at low load, which was consistent with the capacitance change caused by the deformation of the tip of the microcone at a low load. We also tested the LMPS for rapid capacitance changes under different loading pressures, and the results showed that the LMPS could respond to the load very quickly, and the signal was stable under the load condition (Figure 4b). However, there was a certain baseline drift and comparatively obvious hysteresis because of the long recovery process of TPU [12]. The response time of the compression and release of LMPS was ~300 ms and ~400 ms at a load of 250 Pa, which showed that LMPS had a fast response time and was suitable for small load applications because of its high sensitivity at low load. The LMPS had a fast response time and was suitable for small-signal detection because of its high sensitivity at low load (Figure 4c). After testing the basic performance of the LMPS, we further conducted a cyclic compression test to determine its durability and stability. We used 35 kPa periodically to load/unload 4000 cycles rapidly, and intercepted 500~550 s and 3500~3550 s segments, which showed that the LMPS had good stability and the ability to work for a long time (Figure 4d). In order to understand the limitations of the environment in which the sensor was used, we studied the change in capacitance of the sensor at different temperatures and under different loads. The sensor was insensitive to changes in temperature under different loads, which was due to the air between the electrodes and the small change in the dielectric constant of the TPU when the temperature changed. Additionally, in order to understand the performance of the sensor, a hysteresis study was conducted, as shown in Figure 4e. The hysteresis response of the sensor was relatively obvious at low loads of recovery, because the TPU was stiffer, leading to its weak resilience performance. The average value of hysteresis was calculated to be ~10% at 50 kPa. The performance and process of similar microstructured sensors in the literature were provided in Table 1. Figure 4f illustrates a schematic of the experimental setup for the electrical properties of the sensor. The above results illustrate the effectiveness of using laser engraving to fabricate the microstructure on natural wood and wide prospects for LMPS.

### 2.5. Potential Applications for Human Motion Detection

The potential of the LMPS for human motion detection is demonstrated here. As shown in Figure 5a–f, the LMPS is fixed at ear wash ball blowing, the face, the joint of a forefinger, the elbow, and the knee. It can be seen that with airflow, finger flexion, arm bending, knee bending, and facial expressions of cheek-bulging, the capacitance of the LMPS changes quickly and is maintained. When the motion stops and the joint returns to the original position, the LMPS removes the compression, and the capacitance quickly returns to the initial value. The above experiments show that the LMPS exhibited excellent performance in pressure sensing and motion detection, further demonstrating the great potential of the fabrication of LMPS from natural wood in the fields of artificial skin, robotics, and flexible sensing.

## 3. Methods and Materials

### 3.1. Fabrication of the LTE and Microcone Array Pressure Sensor

The main fabrication procedure of the LIG/TPU electrode and microcone array Pressure Sensor (LMPS) is illustrated in Figure 6a. First, the planks (100% pine, ~1 cm of thickness) were covered by an XF-630 flame retardant of nitrogen and phosphorus (Shanghai, China) for five minutes to ensure a shallow penetration depth of flame retardant and dried using a hot air blower. After drying, LTE, a top LIG electrode, and a bottom LIG electrode were scribed and engraved with a semiconductor laser (450 nm wavelength, maximal laser power of 5 W and 60 mm s^−1^ scan speed from DAJA, Dongguan, China). The environment of synthesized LIG and microcone was in the air. For the microstructure, we expected ablation to occur while engraving the microstructure in the air; the ablation produces microstructure groove patterns with deeper depth, which results in a higher microcone height of the dielectric layer and higher sensitivity. However, the oxygenated environment is not conducive to the production of LIG, so we treated the surface of the wood panel with flame-retardant treatment to generate more LIG in the air environment. In our research, LIG came from the laser scribe on the surface of the wood. The top LIG/TPU electrode was obtained by scribing the laser on the planks according to the set square area to form the LIG (Figure 6b), and finally coating the LIG part with the configured TPU solution (TPU+DMF) (the TPU solution was prepared by mixing TPU particles with N-N dimethylformamide in the ratio of 1:3, waiting for 30 min to soften the TPU particles, and then stirring the solution for 30 min to completely mix the TPU particles with N-N dimethylformamide. The solution was left for half an hour for the air bubbles to escape from the solution, and then it could be used as the substrate of electrode). The LIG with the solution was placed in a vacuum drying oven for 30 min to make the TPU solution fully penetrate into the LIG, curing it at 80 °C for 4 h and then peeling it off. Similar to the top LIG/TPU electrode, the bottom LIG electrode with the microhole array was prepared by scribing the square area with laser, then engraving the area according to a set of dot matrices. By coating it with the TPU solution, vacuuming it, curing it at 80 °C for 4 h, and peeling it off, we obtained the bottom LIG/TPU electrode with the microcone array of TPU. We used insulating transparent tape to integrate the top LIG electrode and the bottom LIG electrode with the microcone array.

### 3.2. Performance Test of the LTE and Microcone Array Pressure Sensor

Raman spectra were measured with Raman spectroscopy (WITec, Apyron, Ulm, Germany) equipped with a 532 nm laser and a 785 nm laser, and X-ray photoelectron spectroscopy (XPS) (Thermo Fisher Scientific K-Alpha, Waltham, MA, USA) was performed to evaluate the detailed element information. The thermogravimetric analysis was carried out in the air by NETZSCH STA 449F3 (Bavaria, Germany). A scanning electron microscope (TESCAN MIRA3, Brno, Czech) was used to investigate the surface morphology of the LTE and the microcone of the TPU.

The sheet resistances of the LTE were measured using a Keithley four-point probe meter (Jingge St-2253, Suzhou, China). The sheet resistance of the LTE could be obtained directly from the above machine. A high-precision single-axis electrodynamic force tester (ZQ-990B, Zhiqu Precision Instrument Co., Ltd., Dongguan, China) was used to compress and release the LTE and LMPS repeatedly.

The capacitance of the pressure sensor was measured at room temperature (25 °C) using a desktop digital bridge (GF5000, Guofeng Electronic Technology Co., Ltd., Changzhou, China) with a frequency of 1 kHz, an AC voltage of 3.0 V and a recording interval of 1 s. Here, the sensitivity is defined by:S = (∂(∆C/C_0_)/∂P)
where C and C_0_ stand for the resulting capacitance and the original capacitance, respectively, without the loading pressure (P).

The hysteresis was calculated by:$$hysteresis\left(\%\right)=\left|\frac{\left(\sigma_{L}-\sigma_{U}\right)}{\sigma_{M}}\right|\times 100\%$$
where the subscripts *L*, *U*, and *M* represent the loading, unloading, and maximum values of the stress (σ) at a particular pressure, respectively [52,54].

## 4. Conclusions

In this paper, we reported a simple, efficient, and fast method to deal with the overall flexibility limitations of a sensor caused by metal electrodes and adhesives by transferring LIG to natural wood as well as microstructures with high sensitivity. In addition, the effects of different processing parameters such as laser power, scan speed, defocus distance, and flame-retardant treatment on the sheet resistance of the LTE were explored. The LIG generated by optimizing the laser parameters (50% laser power, 48 mm s^−1^ scan speed, and 0.5 mm defocus distance) yielded a sheet resistance as low as 14.7 Ω/sq. On the basis of this electrode, microstructures were formed on the electrode surface, which further improved the sensitivity of the sensor. The LMPS provides high sensitivity (0.11 kPa^−1^), an ultrawide pressure detection range (20 Pa to 1.4 MPa), a fast response (~300 ms), and good stability (>4000 cycles, at 0–35 kPa). This method allows the flexibility of the sensor to be improved and promotes the further development of flexible electronic devices with LIG from lasers for more potential applications.

## Figures and Tables

**Figure 6 molecules-28-05339-f006:**
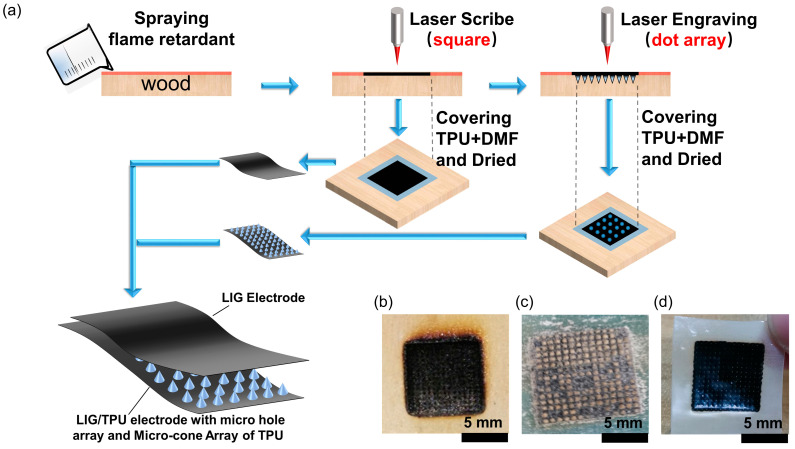
(**a**) Main fabrication procedure of the pressure sensor comprising LTE and an LIG/TPU electrode with microhole array, and dielectric layer of TPU with microcone array. (**b**) An optical image of the surface of the wood after laser engraving. (**c**) An optical image of the LIG/TPU electrode with microhole array, and dielectric layer of TPU with microcone array. (**d**) An optical image of the LIG/TPU electrode with microhole array.

**Figure 1 molecules-28-05339-f001:**
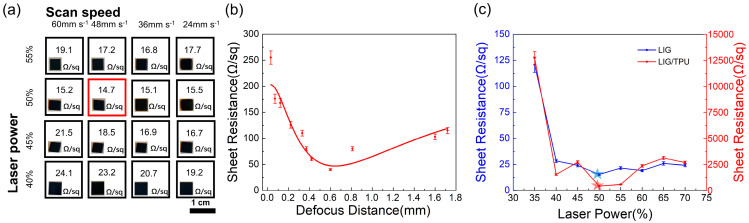
(**a**) LTE was engraved with different laser scan speeds and power, and their sheet resistance and optical photos of corresponding samples were placed in the bottom left corner of every square; the red square is the sample with the minimum surface resistance by optimizing. (**b**) The change in sheet resistance with the defocus distance in the case of 48 mm s^−1^ scan speed and 50% laser power. (**c**) The sheet resistance of LIG and LIG/TPU with different laser power with scan speed 48 mm s^−1^, defocus distance 0.5 mm.

**Figure 2 molecules-28-05339-f002:**
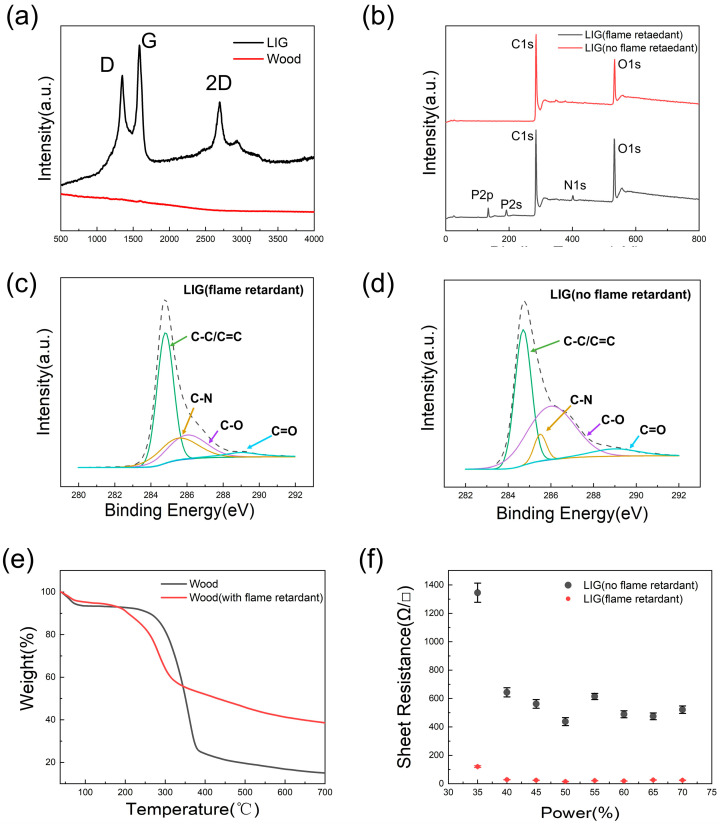
(**a**) Raman spectra of LIG at 532 nm and wood at 785 nm. (**b**) XPS survey spectra of the LIG (flame retardant) and LIG (not flame retardant). (**c**,**d**) The detailed element information of the XPS C1s spectra of the LIG (flame retardant) and LIG (no flame retardant). (**e**) Thermogravimetric curves of wood and wood with flame retardant. (**f**) Sheet resistance of LIG between flame retardant and no flame retardant on different laser powers.

**Figure 3 molecules-28-05339-f003:**
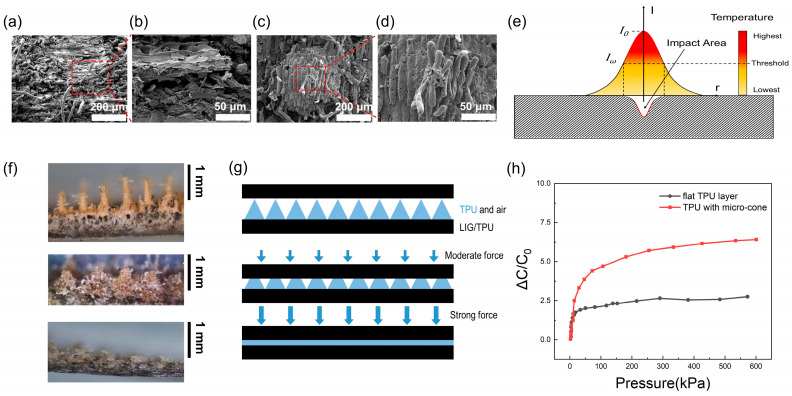
(**a**,**b**) SEM images of the surface of the LTE. (**c**,**d**) SEM images of the surface of the microcone of TPU. (**e**) A schematic of the mechanism of microstructure formation using laser engraving. (**f**) An optical image of microcone fabricated by different laser powers and 48 mm s^−1^ scan speed. (**g**) A schematic of the sensing mechanism for the dielectric layer with microcone. (**h**) Relative capacitance variation of TPU with microcone and flat TPU layer.

**Figure 4 molecules-28-05339-f004:**
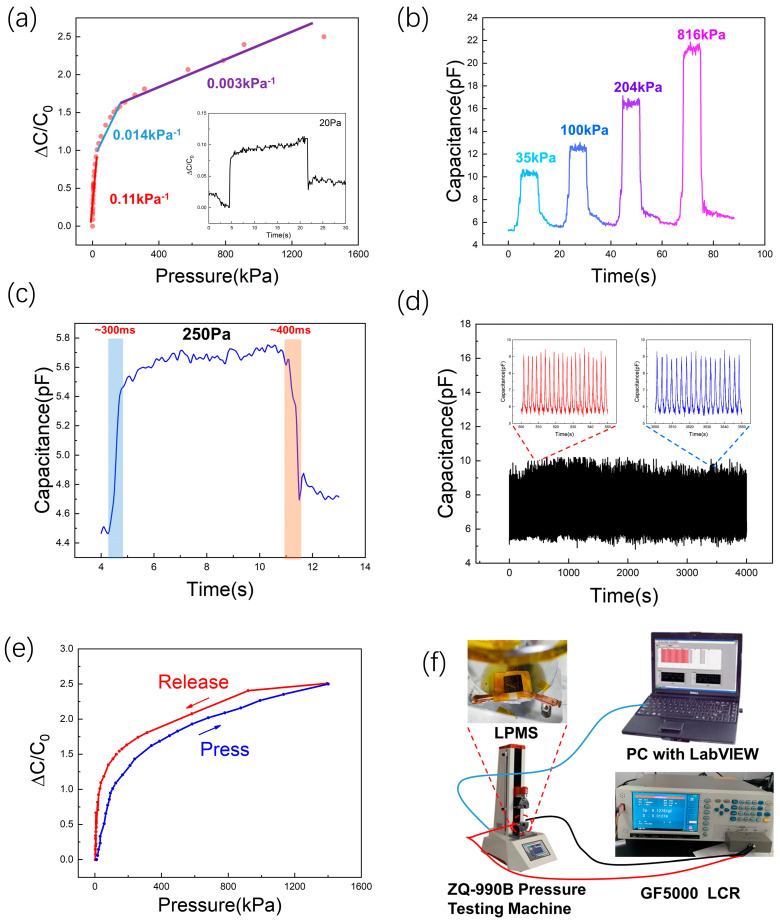
(**a**) Electrical output of the LMPS at different loadings and the minimum detection limit. (**b**) Recognition of different pressure variations of the proposed sensor from low pressure to high pressure. (**c**) The response and recovery speed of the LMPS at a loading of 250 Pa. (**d**) Mechanical durability of the LMPS at a loading of 35 kPa. (**e**) Hysteresis response of the LMPS against pressure. (**f**) A schematic of the experimental setup for the electromechanical characterization of the sensor.

**Figure 5 molecules-28-05339-f005:**
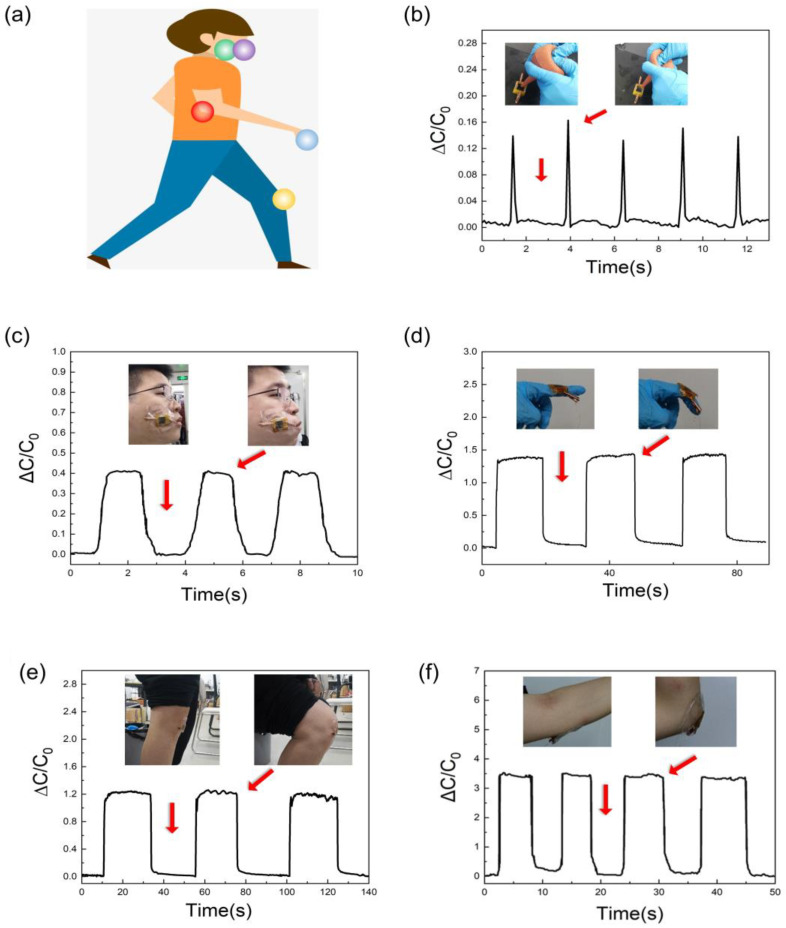
(**a**) Monitoring human motion with the LMPS at different sites. (**b**) Ear wash ball blowing, (**c**) facial expression of cheek-bulging, (**d**) finger movement, (**e**) knee bending, and (**f**) arm bending.

**Table 1 molecules-28-05339-t001:** Table comparing the material, structure, fabrication, highest sensitivity, working range, and response & recovery time of the sensor reported in the present work with similar micro-structured sensors in the literature.

Material	Structure	Fabrication Method	Highest Sensitivity (kPa^−1^)	Working Range (kPa)	Response & Recovery Time (s)	Refs.
CNT/PDMS	gradient micro-dome	micro-engraving	0.065	0–1700	<0.1	[31]
Graphene–PVAc	nanofiber	electrospinning	0.014	2–320	0.4	[52]
rGO-TPU	foam	freeze-drying, dip-coating and chemical reduction	0.0152	20–1940	0.16	[19]
CNT-Ecoflex	foam	dip-coating	1.52	0–50	0.094	[53]
Wood-TPU	micro-cone	laser scribe and engraving	0.11	20–1400	0.3	This work

## Data Availability

Data will be made available on request.

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
