# Peer review of "Flexible Microstructured Capacitive Pressure Sensors Using Laser Engraving and Graphitization from Natural Wood"

_molecules, 2023, doi:10.3390/molecules28145339_

Round 1

Reviewer 1 Report

In the article entitled “Flexible Micro-Structured Capacitive Pressure Sensors by Laser 2 Engraving and Graphitization from Natural Wood”, the authors have presented a flexible capacitive pressure sensor made entirely of thermoplastic polyurethane (TPU) and laser-induced graphene (LIG) derived from wood. The authors conducted several experiments to assess the material morphology, microstructure, chemical properties and electromechanical characterizations to assess the performance of the proposed sensor. The work is interesting and indeed publishable. Though the manuscript treats the burgeoning field of piezocapacitive sensing with great rigor, there are few issues that must be addressed before publishing in final form:

1.      Several reports of LIG based capacitive wearable sensors are there in literature. The novelty of the work must be clearly stated.

2.      Sheet resistance should be expressed in terms of Ω/sq in line 125.

3.      Please comment on the environment of synthesizing LIG. Was air present? How oxidation was prevented?

4.      In general explicitly specify what the error bars represent (for instance Fig. 2 c).

5.      Error bars missing in fig. 2d?

6.      “but wood is not measurable at 532 nm” line 154; kindly provide a reference.

7.      In a work like this, a table comparing several parameters/figure of merit of similar sensors in literature must be provided.

8.      Mechanical characterization experiments to determine compressive modulus, hysteresis, elongation at break, etc are very important for work proposing wearable sensors. The authors may refer to the following articles for inspiration:

a.       D. Sengupta, L. Lu, D. R. Gomes, B. Jayawardhana, Y. Pei, A. G. P. Kottapalli, ACS Appl. Mater. Interfaces 2023, 15, 22351

b.      Chhetry, A.; Kim, J.; Yoon, H.; Park, J. Y. Ultrasensitive Interfacial Capacitive Pressure Sensor Based on a Randomly Distributed Microstructured Iontronic Film for Wearable Applications. ACS Appl. Mater. Interfaces 2019, 11 (3), 3438–3449.

9.      Explain the experimental setup for pressure calibration experiments with a schematic.

Author Response

Dear Reviewer 1:

       We have studied all the comments by carefully and made corresponding revision in the manuscript. Please check the point-to-point responses and kindly offer further consideration.

The main corrections in the manuscript and the responds to the reviewer’s comments are as following attachment file.

Reviewer 2 Report

This work represents a flexible capacitance pressure sensor by LIG and TPU by tuning laser. The idea is interesting but writing and contribution need more improvements. Authors are suggested to revise and response with following comments. Current decision is “Major revision”.

1.      The stability and reproducibility of laser engraving microstructure need to be detail discussed with statics data. In the meantime, the beam profile of 450 nm laser is suggested to provide to clarify the spatial distribution of intensity.

2.      What will be the correlation between laser engraving microstructure to pressure sensing performance? Currently, only correlation to laser condition including scan speed and power is not a good scientific work. Some mechanism and modeling should be given for future improvements.

3.      Due to the capacitive structure of 2 electrodes and micro-cone array, authors have to clearly illustrate to how to make the top electrode and bottom electrode with the conductive path for signal measurement. As shown in Fig. 1(a), LIG electrode with should be top electrode and LIG/TPU with micro cone array? Where is the LIG formed? How to connect the top and bottom electrode? There should be a picture to demonstrate to real situation.

4.      How to author measure sheet resistance? It should be detail presented since these fabricated micro structure is not a uniform film. It is not convinced to use sheet resistance to present its behavior due to uniform thin film is preferred for it by 4-point prober.

5.      The EDS mapping with SEM picture are suggested to let readers understand the distribution of micro cone array.

6.      Some pictures of 3D scanning of microstructure are suggested using VHX-7000, Keyence.

7.      There look some baseline increases shown in Fig. 5(b). Authors are suggested to discuss this recovery performance.

8.      A comparison table and discussion with other flexible pressure sensors are strongly suggested for readers.

9.      Since the capacitance level is very low (e.g., 10-20 pF), the inference from RF signal, temperature, connection, and wire bending could be a big issue. Please authors discuss this part.

10.  Some typos should be corrected as listed below. Authors are strongly suggested to revise all and check in detail.

A.          A space to be put in front of brackets, e.g., line 81- Sensors (LMPS), Fig. 3(e), wood (with flame retardant), Fig. 3(f) LIG (no flam retardant), etc.

B.          A space to be put in form of unit, e.g., line 81- 1 cm, line 85- 450 nm, line 131- 0.5 mm,

C.          What is DMF shown only in Fig. 1(a) without explanation in content?

D.         Micro-cone array and micro hole array should be clearly drawn in Fig. 1(a)

E.          Line 125, unit of sheet resistance is not correct.

Author Response

Dear Reviewer 2:

       We have studied all the comments by carefully and made corresponding revision in the manuscript. Please check the point-to-point responses and kindly offer further consideration.

The main corrections in the manuscript and the responds to the reviewer’s comments are as following attachment file.

Round 2

Reviewer 1 Report

The authors have satisfactorily replied to all the issues that were raised previously. 

Minor editing required. 

Reviewer 2 Report

The quality of revised manuscript is ready for publication.
